

**Bryophyte-dominated biological soil crusts mitigate soil**
**erosion in an early successional Chinese subtropical forest**
Steffen Seitz[1], Martin Nebel[2,3], Philipp Goebes[1], Kathrin Käppeler[1], Karsten Schmidt[1],
Zhengshan Song[1], Carla L. Webber[4], Bettina Weber[5], Thomas Scholten[1]
[1] Department of Geosciences, Soil Science and Geomorphology, University of Tübingen, Tübingen, 72070,
Germany
[2] State Museum of Natural History, Stuttgart, 70191, Germany
[3] Nees Institute for Biodiversity of Plants, University of Bonn, Bonn, 53115, Germany
[4] Department of Geosciences, Federal University of Rio Grande do Sul, Porto Alegre, 90040-060, Brazil
[5] Multiphase Chemistry Department, Max Planck Institute for Chemistry, Mainz, 55128, Germany
*Correspondence to:* Steffen Seitz (steffen.seitz@uni-tuebingen.de)




**Abstract.** This study investigated the development of biological soil crust (biocrust) covers in an early
successional subtropical forest ecosystem and their impact on soil erosion. Within a biodiversity and ecosystem
functioning experiment in Southeast China (BEF China), sediment discharge and runoff measurements were
conducted with micro-scale runoff plots under natural rainfall and biocrust covers were surveyed over a five-year
period.
Results showed that biocrusts occurred widely in our experimental forest ecosystem and developed from initial
light cyanobacteria- and algae-dominated crusts to later-stage bryophyte-dominated crusts in only three years.
Biocrust covers were still increasing after six years of tree growth. Within later stage crusts, 25 bryophyte species
were determined. The development of biocrusts was significantly influenced by the surrounding vegetation cover
and terrain attributes. Besides high crown cover and leaf area index, the development of biocrusts was favoured
by low slope gradients, slope orientations towards the incident sunlight and the altitude of the research plots. Our
measurements showed, that bryophyte-dominated biocrusts were importantly decreasing soil erosion and more
effective in erosion reduction than abiotic soil surface covers. Hence, their significant role to mitigate sediment
discharge and runoff generation in mesic forest environments and their ability to quickly colonize gaps in higher
vegetation layers are of particular interest for soil erosion control in early stage forest plantations. A detailed record
of different biocrust species and their functional influence on soil erosion processes as well as a thorough
monitoring of biocrust covers under closing tree canopy in subtropical forests is required in further studies.

















## 1 Introduction

Biological soil crusts (hereinafter referred to as biocrusts) are a living soil cover, which plays significant functional roles in many environments (Weber et al., 2016). In initial ecosystems, communities of cyanobacteria, algae, fungi, lichens, bryophytes and bacteria in varying combinations are the first to colonize the substrate (Evans and Johansen, 1999). Those highly specialized communities form a biological crust immediately on top or within the first millimetres of the soil surface (Büdel, 2005). Biocrusts generally occur under harsh conditions of temperature or light and when the cover of vascular vegetation is sparse (Allen, 2010). Therefore, biocrusts are generally widespread under dryland conditions (Berkeley et al., 2005; Belnap, 2006; Büdel et al., 2009), whereas under mesic conditions they mostly occur as a successional stage after disturbance or in environments under regularly disturbed regimes (Büdel et al., 2014).

Biocrusts are generally less capable of competing with phanerogamic plants and thus the crust development is limited when closed plant canopies or litter layers come into play (Belnap et al., 2003a). This limitation is due to the competition for light (Malam Issa et al., 1999) and nutrients (Harper and Belnap, 2001). Nevertheless, a high number of studies could not clarify the relation between biocrust cover and vascular plant cover and some studies even showed a positive correlation (Belnap et al., 2003b). This coherence was explained with enhanced nutrient levels provided for vascular plants growing on crusted compared to non-crusted soil surfaces (Kleiner and Harper, 1977; Belnap, 2002). Therefore, the improvement of soil fertility by biocrusts has been shown to be fundamental for the development of vascular plant communities in some regions (St. Clair and Johansen, 1993; Harper and Belnap, 2001). Biocrusts are able to quickly colonize gaps in higher vegetation layers (Belnap et al., 2003a) or gaps appearing after disturbance (Dojani et al., 2011; Chiquoine et al., 2016). Such disturbances can occur e.g. in forest ecosystems by treefall or after clear-cutting (Barnes and Spurr, 1998). The complete removal of a forest signifies a harsh cut in vegetation development and creates a starting point for new vascular plant as well as biocrust communities (Bormann et al., 1968; Keenan and Kimmins, 1993; Beck et al., 2008). Nevertheless, the development of biocrusts in early successional forest ecosystems under a closing tree canopy has not been in focus of research so far (Su et al., 2007; Zhang et al., 2016). Furthermore, evidence for different biocrust types in mesic vegetation zones and especially from southeast Asia are rare (Büdel, 2003a; Bowker et al., 2016).

Functions of biocrusts have been investigated for decades, but less attention has been paid to their spatial distribution and characteristics (Allen, 2010). Biocrust cover varies across different scales (from centimetres to kilometres) and depends not only on the surrounding vascular vegetation cover, but also on geomorphology and (micro-)topography or terrain (Evans and Johansen, 1999; Ullmann and Büdel, 2003; Kidron et al., 2009; Bowker et al., 2016). Different biocrust distributions could be related to elevation and terrain-influenced microclimatic gradients (Kutiel et al., 1998), different geomorphic zones (Eldridge, 1999), varying aspects (George et al., 2000) or soil types (Bu et al., 2016). To our knowledge, investigations on the influence of small-scale (centimetres to metres) topographic variations on biocrust development are rare and further studies will assist in the understanding of their abundance and distribution (Garcia-Pichel and Belnap, 2003; Bu et al., 2016; Bowker et al., 2016). Furthermore, as the development of biocrusts is characterized by a high complexity and spatial heterogeneity with many micro-climatic and micro-environmental factors, it is of great significance to conduct comparative studies on the spatial distribution of biocrusts (Bu et al., 2013b). This is particularly true for initial forest ecosystems (Weber et al., 2016).



Moreover, biocrusts have been recognized as having major influences on terrestrial ecosystems (Buscot and Varma, 2005), as they protect soil surfaces against erosive forces by both wind and water (Bowker et al., 2008; Zhao et al., 2014), enhance soil stability (Malam Issa et al., 2001; Warren, 2003) and influence the hydrological cycle (Belnap, 2006). Nevertheless, impacts of biocrusts on soil stability and soil hydrology may differ with regard to soil texture, surface roughness, water repellency and finally different crust species and developmental stages (Warren, 2003; Belnap and Büdel, 2016). Furthermore, studies that directly relate biocrust cover to rates of soil erosion are few (Allen, 2010) and the influence of biocrusts on sediment discharge and runoff in humid climates has been largely disregarded (Belnap and Lange, 2003; Weber et al., 2016).

This study aims to investigate the development of biocrust cover in an early successional subtropical forest ecosystem and the impact of those biocrusts on soil erosion. Therefore, interrill erosion was measured with runoff plots and the occurrence, distribution and development of biocrusts was recorded. The study was conducted in an experimental forest plantation (BEF China), which aims to study biodiversity and ecosystem functioning relationships in southeast China (Yang et al., 2013; Bruelheide et al., 2014). The following hypotheses were addressed:

(1) Biocrusts widely develop in early successional subtropical forest ecosystems, but crust cover decreases with ongoing tree growth.

(2) The development of biocrusts is influenced by the surrounding vegetation cover, but also by soil and terrain attributes.

(3) Biocrusts mitigate interrill soil erosion in early successional subtropical forest ecosystems.

## 2 Material and methods

2.1 Study site and experimental design

The study was carried out within the BEF China experiment (Bruelheide et al., 2014) in Xingangshan, Jiangxi Province, PR China (29°06.450′ N and 117°55.450′ E). The experimental area is located in a mountainous landscape at an elevation of 100 m a.s.l. to 265 m a.s.l. with slopes from 15° to 41° (Scholten et al., 2017). The bedrock is non-calcareous slates weathered to saprolite and predominant soil types are Cambisols with Anthrosols in downslope positions and Gleysols in valleys (Scholten et al., 2017). The mean annual temperature is 17.4 °C and the annual precipitation is 1635 mm with about 50 % falling during May to August (Goebes et al., 2015). The climate is typical for summer monsoon subtropical regions. The potential natural vegetation of this region is a subtropical broadleaved forest with dominating evergreen species. The experimental area is structured in 566 research plots (25.8 m × 25.8 m each) at two sites (A and B) which were clear-cut and replanted with 400 tree saplings per plot in 2009 and 2010, respectively (Yang et al., 2013). A selection of 34 research plots (VIPs, Very Intensively studied Plots) was used for this study.

2.2 Field methods





Biocrust cover was determined photogrammetrically in 70 selected runoff plots (ROPs, 0.4 m × 0.4 m; Seitz et al.,
2015) at five timesteps (November 2011, May 2012, May 2013, May 2014 and May 2015) and general biocrust
types were described in the field. During the rainy season in summer 2013, an extended survey linked to soil
erosion measurements was conducted in 170 ROPs (see below and Table 1). At each ROP, perpendicular images
were taken with a single lens reflex camera system (Canon 350D, Tokio, Japan) and processed with the grid
quadrat method in GIMP 2.8 using a digital grid overlay with 100 subdivisions (cf. Belnap et al., 2001). Stone
cover and biocrust cover were separated by hue distinction. A continuous leaf litter cover, which may impede
analyses, was not present during measurements. Biocrusts were collected in 2013 and samples were dried at 40 °C
(Dörrex drying unit, Netstal, Switzerland). The identification of species was carried out by morphological
characteristics using a stereomicroscope (Leitz TS, Wetzlar, Germany), a transmitted-light microscope (Leitz
Laborlux S, Wetzlar, Germany) and ultraviolet light. Bryophytes (dominating taxa in 2013) were determined to
the species level, wherever possible and separated into mosses (Bischler-Causse, 1989; Moos flora of China: Gao
et al., 1999; 2001; 2002; 2003; 2005; 2007; 2008; 2011) and liverworts (Zhu, 2006; Söderström et al., 2016 and
Alfons Schäfer-Verwimp, personal communication). Comparisons were conducted with specimen hosted in the
herbarium of the State Museum of Natural History in Stuttgart, Germany (Herbarium STU).
Sediment discharge and surface runoff were measured within 34 research plots on five micro-scale ROPs each in
summer 2013 (n=170, Table 1). After four timesteps, 334 valid ROP measurements entered the analysis (for
detailed information see Seitz et al., 2016). Sediment discharge was sampled, dried at 40 °C and weighed, whereas
surface runoff and rainfall amount were measured in situ. At every ROP, crown cover and leaf area index (LAI)
were measured with a fish-eye camera system (Nikon D100 with Nikon AF G DX 180°, Tokio, Japan) and
calculated with HemiView V.8 (Delta-T devices, Cambridge, UK). Measurements of tree height and crown width
were provided by Li et al. (2014) at research plot scale (n=34). Tree species richness and tree composition resulted
from the experimental setup of BEF China (Bruelheide et al., 2014).
Soil attributes (Table 1) were determined for every research plot (n=34) using pooled samples from nine point
measurements each. Soil pH was measured in KCl (WTW pH-meter with Sentix electrodes, Weilheim, Germany),
bulk soil density was determined by the mass-per-volume method and total organic carbon (TOC) was measured
using heat combustion (Elementar Vario EL III, Hanau, Germany). Soil organic matter (SOM) was calculated by
multiplying TOC with the factor 2 (Pribyl, 2010).
2.3 Digital terrain analysis
Terrain attributes (Table 1) were derived from a digital elevation model (DEM, Scholten et al., 2017) at research
plot scale (n=34). Attributes were the terrain ruggedness index (TRI, Riley et al., 1999) to describe the
heterogeneity of the terrain, the Monte-Carlo based flow accumulation (MCCA, Behrens et al., 2008) to diagnose
terrain driven water availability, altitude above sea level to address elevation effects and the eastness and the
northness (Roberts, 1986) to describe plant related climatic conditions. Those terrain attributes cover major
landscape features of the experimental area and were not correlated. Slope was additionally measured with an
inclinometer at every ROP (n=170, see Seitz et al., 2016).





**[ Table 1 ]**

2.4 Statistical methods
The temporal development of biocrust covers (1) from 2011 to 2015 was assessed at five timesteps within 70 ROPs
(see above) by an analysis of variance (ANOVA) and Tukey's Honestly Significant Difference (HSD) test (n=350).
The influences of vegetation, soil and topographic attributes on biocrust cover (2) in 170 ROPs (see above) were
assessed by linear mixed effects (LME) models (n=334). Crown cover, bulk soil density, SOM, pH, altitude, slope,
MCCA, TRI, eastness, northness and tree species richness were fitted as fixed effects and biocrust cover as
response variable. The attributes were tested with Pearson's correlation coefficient before fitting. LAI was fitted
individually in exchange to crown cover due to multi-collinearity. Experimental site and research plot were fitted
as random effects and hypotheses were tested with an ANOVA type 1 with Satterthwaite approximation for
degrees of freedom.
The influences on soil erosion (3) were assessed by LME models with restricted maximum likelihood (n=334) and
sediment discharge and surface runoff as response variables, respectively. Crown cover, slope, surface cover,
SOM, rainfall amount and tree species richness were fitted as fixed effects. Surface cover was than split into
surface cover by biocrusts and by stones, which entered the analysis as fixed conjoined factors. Precipitation events
nested in plot, tree species composition, experimental site and ROP nested in plot were fitted as random effects.
Attributes were not correlated. The hypothesis was tested with an ANOVA type 1 with Satterthwaite
approximation for degrees of freedom. Moreover, the Wilcoxon rank sum test was applied to test for differences
between biocrust cover and stone cover on sediment discharge and surface runoff. Therefore, the dataset was split
into data points where biocrust cover exceeds stone cover (n=281) and data points where stone cover exceeds
biocrust cover (n=53).
All response variables were log-transformed before modelling and analyses were performed with R 3.1.2 (R Core
Team, 2014). LME modelling was conducted with "lmerTest" (Kuznetsova et al., 2014) and rank sum tests with
"exactRankTests" (Hothorn and Hornik, 2015). Figures were designed with "ggplot2" (Wickham, 2009).

**3 Results**
3.1 Temporal development of biocrust cover
Biocrusts were detected in 94 % of all ROPs and their cover within ROPs ranged between 1 % and 88 % over all
five years. The mean biocrust cover of all ROPs more than tripled from their installation in 2011 to the last
measurement in 2015 (Fig. 1). This increase was significant from 2011 to 2015 and from 2012 to 2013, 2013 to
2014 and 2014 to 2015 (p<0.001).

**[ Figure 1 ]**



Whereas a clear bryophyte-dominance of biocrusts was evident at the time of sampling in 2013, different
successional stages were identified in the field and on ROP photos from 2011 to 2015 (Fig. 2). In 2011, a smooth,
light cyanobacteria- and algae-dominated crust with traces of lichens and few bryophytes indicated a pioneer stage
of biocrust development (cf. Colesie et al., 2016). In 2013, 25 moss and liverwort species were classified (Table
2) and formed a bryophyte-dominated crust with cyanobacteria, algae, lichens and micro-fungi still observed in
minor numbers within ROPs. The same was true in 2015, but first evidence of vascular plants (*Selaginella* and
*Poaceae*) indicated a further advanced biocrust stage.

**[ Figure 2 ]**

**[ Table 2 ]**

3.2 The influence of vegetation, soil and terrain on biocrust cover
The development of biocrust cover in 2013 was positively influenced by crown cover and LAI as attributes of the
surrounding vegetation (Table 3). Furthermore, it was negatively affected by slope and northness and slightly
positively affected by the altitude of the research plots (Table 3). Further soil or terrain attributes did not affect
biocrust cover.

**[ Table 3 ]**

3.3 The impact of biocrust cover on soil erosion
Both biocrust and stone cover, as well as soil surface cover (comprising both biocrust and stone cover) negatively
affected sediment discharge (p<0.001, Table 4). In addition, soil surface cover negatively affected surface runoff
(p=0.003). However, only biocrust but not stone cover mediated the effect of runoff. Furthermore, crown cover,
SOM and rainfall amount affected sediment discharge, whereas runoff was affected by crown cover and rainfall
amount. ROPs primarily influenced by stone cover showed higher sediment discharge and surface runoff than
those primarily influenced by biocrust cover (Fig. 3).

**[ Table 4 ]**





**[ Figure 3 ]**

**4 Discussion**
4.1 Temporal development of biocrust cover
Biocrusts were detected widely within the experiment and occupied a considerable area in the interspaces of the
growing tree community. Thus, the first part of hypothesis 1 can be confirmed, as biocrusts colonized the newly
created habitats originating from the disturbance by forest clear-cutting and weeding (Bruelheide et al., 2014). The
deforestation provided a local arid micro-environment which initiated early biocrust development (Büdel, 2003b).
At this early stage of the ecosystem, biocrusts were highly competitive and formed a pioneer vegetation (Langhans
et al., 2009), which then prepared the upper soil layer for further growth of vascular plants by the input of carbon
and nitrogen (West, 1990; Evans and Johansen, 1999). Biocrusts generally facilitate the succession of vascular
plants to more advanced stages (Bowker, 2007). Accordingly, tree growth provide shade and protection from wind,
which then leads to advancement in biocrust development. The bryophyte-dominance of biocrusts in 2013
documented this development into a later and somewhat moister successional stage (Williams and Büdel, 2012).
Biocrusts are often dominated by one organism group, with cyanobacterial crusts being indicators for early stage
crusts and drier conditions (Malam Issa et al., 1999; Malam Issa et al., 2007). The successional development of
biocrusts within the BEF China experiment seemed to be faster than e.g. reported by Zhao et al. (2010) for the
Chinese Loess Plateau, who claimed biocrusts from a 3-year old site as early successional dominated by
cyanobacteria. Bryophytes in biocrusts have received comparatively little attention and in Asia only 23 different
species have been reported up to now (Seppelt et al., 2016). Thus, this study with 25 recorded moss and liverwort
species, most of them being new records within Asian biocrusts (Burkhard Büdel, personal communication)
substantially increases the knowledge on biocrusts of this region.
Nevertheless, the extent of biocrusts was strongly increasing since 2012 i.e. three years after tree replantation and
still gaining coverage in the sixth year after our experimental setup. Thus, the second part of hypothesis 1 has to
be rejected. Even if single trees were already up to 7.4 m high (Li et al., 2014) and LAI was up to 5.35 in 2013,
biocrusts still remained competitive within the forest ecosystem. Moreover, increasing crown cover and LAI
seemed to foster the development of bryophyte-dominated biocrusts at this ecological stage. It is assumed that
with continuing tree growth the biocrust communities will adapt and the composition of moss and liverwort species
will further change (Eldridge and Tozer, 1997). Thus, bryophytes will likely switch from species favouring sunny
habitats to more shade-tolerant species (Zhao et al., 2010; Müller et al., 2016). In addition, there might also be a
reduction in bryophyte diversity due to shady conditions, where only smaller number of species could prevail.
Nevertheless, adapting biocrusts seem to be able to coexist widely with vascular plants under a nearly closed tree
canopy, even if it is assumed that biocrust cover will decrease in later years with an increasing leaf litter layer
(Belnap and Lange, 2003). In this context, the ecological roles of biocrusts in succession models and plant
restoration are of interest (Hawkes, 2004; Bowker, 2007). Restoration of biocrusts in disturbed ecosystems could
be a practical approach to improve and accelerate plant community rehabilitation after disturbance and there are
several projects under way to establish successful restoration techniques (Rosentreter et al., 2003; Bowker, 2007;
Chiquoine et al., 2016).





4.2 The influence of vegetation, soil and terrain on biocrust cover

The development of biocrusts was influenced by vegetation and terrain attributes, but not by soil attributes. Thus, hypothesis 2 can be partly confirmed. As already shown before, high crown cover and LAI affected the development of biocrust cover in 2013. This finding is due to the successional alteration of biocrusts towards bryophyte-dominance. Mosses and liverworts profit from humid conditions and a higher protection from light compared to cyanobacteria- or lichen-dominated crusts (Ponzetti and McCune, 2001; Marsh et al., 2006; Williams et al., 2013). Environmental factors such as water content, light intensity and temperature influence e.g. photosynthesis and respiration (Zhao et al., 2010; Weber et al., 2012).

Furthermore, several terrain attributes affected biocrust cover. Slope was the most prominent of those factors, causing a considerable decline in biocrust cover with increasing slope, being explained by their decreasing ability to fasten themselves on the soil surface at high slope angles, especially when large surface water flows occurred during rainfall events (Chamizo et al., 2013; Bu et al., 2016). Northness showed a positive impact on biocrust covers and indicated that slope orientations towards the incident sunlight directly influence the biocrust development (Bowker et al., 2002; Zaady et al., 2007). Furthermore, biocrust development depended on the altitude, probably by affecting microclimatic conditions (Kutiel et al., 1998; Chamizo et al., 2016; Bu et al., 2016).

Interestingly, SOM and pH did not affect biocrust cover in this study. Increased organic matter contents and acidic conditions, as they were found at the experimental area (Scholten et al., 2017), generally favour the development of bryophyte-dominated biocrusts (Eldridge and Tozer, 1997; Seppelt et al., 2016). Nevertheless, discrepancies between the research plots were small and apparently not large enough to cause prominent differences in biocrust development.

4.3 The impact of biocrust cover on soil erosion

Biocrust cover clearly mitigated interrill soil erosion in this early stage ecosystem and thus hypothesis 3 was confirmed. Biocrusts attenuate the impact of raindrops on the soil surface and greatly improve its resistance against soil erosion (Eldridge and Greene, 1994; Goebes et al., 2014; Zhao et al., 2014). Moreover, they have the ability to glue loose soil particles by polysaccharides extruded by cyanaobacteria and green algae (Buscot and Varma, 2005). In the current study, protonema and rhizoids of mosses and liverworts were observed to be most effective by weaving and thus fixing the first millimetres of the top soil (Bowker et al., 2008). *Pogonatum inflexum* and *Atrichum subserratum* are well known to have a positive effect on erosion control due to their sustained protonema system. Furthermore, bryophytes increase the formation of humus, which in turn assists to bind primary particles into aggregates (Scheffer et al., 2010; Zhang et al., 2016). Thus, biocrusts contribute to the aggregation of soil particles and stabilize the upper soil surface. Furthermore, biocrusts are known to influence hydrological processes such as surface runoff and infiltration rates (Belnap, 2006; Cantón et al., 2011; Chamizo et al., 2012). Just recently, Chamizo et al. (2016) showed that runoff and infiltration also depend on the investigation scale. Whereas point based measurements showed both increasing and decreasing runoff through biocrusts depending on the study site, studies of larger scale (>2 m$^2$) revealed that biocrusts decrease runoff generation (Chamizo et al., 2016). Moreover, reducing effects on runoff are related to the biocrusts species composition (Belnap and Lange, 2003). Especially bryophyte-dominated crusts appear to enhance infiltration and reduce runoff due to their rhizome system, while soil erosion rates stay low (Warren, 2003; Yair et al., 2011). Field studies in Utah, USA, revealed that later stage





biocrusts, containing both lichens and mosses, offer more protection against soil erosion than cyanobacterial crusts
(Belnap and Gillette, 1997). They provide higher infiltration than biocrusts dominated by cyanobacteria (Kidron,
1995) and decrease the aggradation of soil pores by reducing the kinetic energy of raindrops (Eldridge and Greene,
1994). Moreover, biocrusts dominated by bryophytes increase surface roughness and thus slow down runoff
(Kidron et al., 1999; Rodríguez-Caballero et al., 2012). Furthermore, they absorb water and provide a certain water
storage capacity (Warren, 2003; Belnap, 2006). Especially *Leucobryum juniperoideum* is known for its water
absorbing capacity. Whereas a partial stone cover does not enhance infiltration, bryophyte-dominated biocrusts
positively influence the hydrological processes in the top soil layer regarding erosion control. Thus, they actively
mitigate initial soil erosion compared to abiotic components such as stones. This study showed, that biocrust covers
play an important role in the avoidance of severe soil erosion in early successional forest plantations. This effect
should be considered for the replantation of forests in regions endangered by soil erosion. Furthermore, the
artificial cultivation of mosses in such initial forest ecosystems could improve erosion control (Bu et al., 2013a;
Zhao et al., 2016). At this point, the importance of biocrusts in the rehabilitation of disturbed ecosystems comes
into focus again (Bowker, 2007).

**5 Conclusion**
This study investigated the development and distribution of biocrusts in an early stage subtropical forest ecosystem
as well as their impact on interrill soil erosion. The following conclusions were obtained:
(1) Biocrusts occurred widely in this early successional forest ecosystem in subtropical China. They developed
quickly to later-stages in this mesic environment and were already dominated by bryophytes after three years of
tree growth (25 bryophyte species classified). After six years of continuing canopy closure, biocrust cover was
still increasing. Further monitoring under closing tree canopy is of importance to detect changes in biocrust cover
and species composition in subtropical environments.
(2) The surrounding vegetation and underlying terrain affected biocrust development, whereas soil attributes did
not have an effect. Besides high crown cover and LAI, the development of biocrusts was favoured by low slope
gradients, slope orientations towards the incident sunlight and altitude. Further research appears to be necessary
to explain effects of terrain attributes such as aspect or elevation and effects of underlying soils and substrates.
(3) Soil surface cover of biocrusts largely affected soil erosion control in this early stage of the forest ecosystem.
Bryophyte-dominated crusts had erosion-reducing characteristics regarding both sediment discharge and surface
runoff. Furthermore, they were more effective to decrease soil loss compared to abiotic soil surface covers. These
functional properties of bryophytes with regard to soil erosion need to be considered for management practices in
early stage forest plantations. Further research is required on functional mechanisms of different biocrust species
and their impact on soil erosion processes.

**Data availability**
Data are publicly accessible and archived at the BEF China data portal (http://china.befdata.biow.uni-leipzig.de).



**Author contribution**
Steffen Seitz and Thomas Scholten designed the experiment and Steffen Seitz, Zhengshan Song, Kathrin Käppeler
and Carla L. Webber carried it out. Martin Nebel and Kathrin Käppeler classified biocrust types and determined
bryophyte species. Steffen Seitz, Philipp Goebes and Karsten Schmidt performed the statistical models. Steffen
Seitz and Bettina Weber prepared the manuscript with contributions from all co-authors. The authors declare that
they have no conflict of interest.
**Acknowledgements**
We are grateful to the BEF China research group and especially to our students Mario Ahner, Milan Daus, Marlena
Hall, Madeleine Janker, Paula Kersten, Vera Müller and Andrea Wadenstorfer for assistance in fieldwork. We also
thank Alfons Schäfer-Verwimp for assistance in determination of bryophytes, Karl Forchhammer for giving us
first insights into the world of cyanobacteria and the participants of BioCrust3 for helpful comments on the results.
This work was funded by the German Research Foundation (DFG FOR 891/2 and 891/3). We also benefitted from
travel grants by the Sino-German Centre for Research Promotion (GZ 699 and GZ 785).

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





**Tables**
**Table 1: Erosion, soil, soil cover, vegetation and terrain attributes in runoff plots (ROPs, n=170) and on research plots**
**(n=34) in Xingangshan, Jiangxi Province, PR China in 2013.**

|  | *Min* | *Mean* | *Max* |
|---|---|---|---|
| *Runoff plots (ROPs, n=170)* | | | |
| Sediment discharge [g] | 21.6 | 195.5 | 989.0 |
| Surface runoff [ml] | 3.1 | 40.3 | 111.8 |
| Rainfall amount [ml] | 25 | 94 | 178 |
| Slope [°] | 5 | 29 | 60 |
| Soil cover [%] | 0 | 19 | 62 |
| - Biological soil crust cover [%] | 0 | 24 | 62 |
| - Stone cover [%] | 0 | 4 | 42 |
| Crown cover [%] | 0.00 | 0.32 | 1.00 |
| Leaf area index (LAI) | 0.00 | 0.73 | 5.35 |
| *Research plots (n=34)* | | | |
| Bulk soil density [g cm$^{-2}$] | 0.83 | 0.98 | 1.12 |
| Soil organic matter [%] | 4.2 | 6.5 | 9.7 |
| pH (KCl) | 3.24 | 3.66 | 4.00 |
| Altitude [m] | 119 | 167 | 244 |
| MCCA | 0.98 | 2.07 | 3.81 |
| TRI | 0.72 | 2.39 | 3.86 |
| Eastness | -0.86 | 0.09 | 0.99 |
| Northness | -0.87 | 0.23 | 0.99 |
| Tree height [m] | 1.0 | 2.2 | 7.4 |
| Crown width [m] | 0.4 | 1.2 | 3.0 |

**Soil cover: proportion of soil surface area covered by biocrusts or stones, crown cover: proportion of soil surface area**
**covered by crowns of live trees, leaf area index: one-sided green leaf area per unit soil surface area, MCCA: Monte-**
**Carlo based flow accumulation (Behrens), TRI: terrain ruggedness index (Riley), Eastness and Northness: state of**
**being east or north (Roberts), tree height: distance from stem base to apical meristem, crown width: length of longest**
**spread from edge to edge across the crown**





**Table 2: Liverwort and moss species sampled in the BEF China experiment in Xingangshan, Jiangxi Province, PR**
**China in 2013.**

| Family | Species | | Author |
|---|---|---|---|
| Liverworts | | | |
| *Calypogeiaceae* | *Calypogeia* | *fissa* | (L.) Raddi |
| *Conocephalaceaes* | *Conocephallum* | *salebrosum* | Szweyk., Buczk. et Odrzyk. |
| *Lophocoleaceae* | *Heteroscyphus* | *zollingeri* | (Gottsche) Schiffn. |
| *Marchantiacea* | *Marchantia* | *emarginata* | Reinw., Blume et Nees |
| *Acrobolbaceae* | *Notoscyphus* | *lutescens* | (Lehm. et Lindenb.) Mitt. |
| Mosses | | | |
| *Polytrichaceae* | *Atrichum* | *subserratum* | (Harv. et Hook. f.) Mitt. |
| *Pottiaceae* | *Barbula* | *unguiculata* | Hedw. |
| *Bryaceae* | *Bryum* | *argenteum* | Hedw. |
| *Leucobryaceae* | *Campylopus* | *atrovirens* | De Not. |
| *Dicranellaceae* | *Dicranella* | *heteromalla* | (Hedw.) Schimp. |
| *Pottiaceae* | *Didymodon* | *constrictus* | (Mitt.) K. Saito |
| *Pottiaceae* | *Didymodon* | *ditrichoides* | (Broth.) X.J. Li et S. He |
| *Ditrichaceae* | *Ditrichum* | *pallidum* | (Hedw.) Hampe |
| *Entodontaceae* | *Entodon* | spec. | sterile |
| *Hypnacaea* | *Hypnum* | *cupressiforme* | Hedw. |
| *Hypnacaea* | *Hypnum* | *macrogynum* | Besch. |
| *Leucobryaceae* | *Leucobryum* | *juniperoideum* | (Brid.) Müll. Hal. |
| *Bartramiaceae* | *Philonotis* | *marchica* | (Hedw.) Brid. |
| *Bartramiaceae* | *Philonotis* | *mollis* | (Dozy et Molk.) Mitt. |
| *Bartramiaceae* | *Philonotis* | *roylei* | (Hook. f.) Mitt. |
| *Mniaceae* | *Plagiomnium* | *acutum* | (Lindb.) T.J. Kop. |
| *Polytrichaceae* | *Pogonatum* | *inflexum* | (Lindb.) Sande Lac. |
| *Thuidiaceae* | *Thuidium* | *glaucinoides* | Broth. |
| *Mniaceae* | *Trachycystis* | *microphylla* | (Dozy et Molk.) Lindb. |
| *Pottiaceae* | *Trichostomum* | *crispulum* | Bruch |








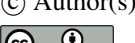


**Table 3: Results of the final linear mixed effects (LME) model for vegetation, soil and terrain attributes on biological**
**soil crust cover in Xingangshan, Jiangxi Province, PR China in 2013 (\*\*\*: p < 0:001; \*\*: p < 0:01; \*: p < 0:05; .: p <**
**0:1; ns: not significant; n=215).**

| | Biological soil crust cover | | | |
|---|---|---|---|---|
| | denDF | F | Pr | estim. |
| *Fixed effects* | | | | |
| Crown cover | 136 | 12.9 | *** | 10.8 |
| Bulk soil density | 37 | 0.03 | ns | 3.65 |
| SOM | 39 | 1.11 | ns | (-)0.95 |
| pH (KCl) | 38 | 2.47 | ns | (-)16.7 |
| Altitude | 37 | 3.69 | . | 0.80 |
| Slope | 191 | 7.53 | ** | (-)2.72 |
| MCCA | 39 | 0.02 | ns | 0.33 |
| TRI | 38 | 0.04 | ns | (-)0.40 |
| Eastness | 37 | 2.73 | ns | (-)4.23 |
| Northness | 37 | 9.14 | ** | 5.99 |
| Tree species richness | 38 | 1.22 | ns | (-)0.27 |
| | | | | |
| *Random effects* | | *SD* | *Variance* | |
| Site | | <0.01 | <0.01 | |
| Plot | | <0.01 | <0.01 | |
| | | | | |
| *Vegetation attribute fitted in exchange to crown cover* | | | | |
| Leaf area index | 107 | 42.8 | *** | 5.98 |

**SOM: soil organic matter; MCCA: monte carlo based flow accumulation; TRI: topographic roughness index; denDF:**
**denominator degrees of freedom; F: F value; Pr: significance; estim.: estimates**










**Table 4: Results of the final linear mixed effects (LME) models for sediment discharge and surface runoff with**
**surface cover split into biological soil crust cover and stone cover in Xingangshan, Jiangxi Province, PR China in 2013**
**(\*\*\*: p < 0:001; \*\*: p < 0:01; \*: p < 0:05; .: p < 0:1; ns: not significant; n=334).**

| | Sediment discharge | | | | Surface runoff | | | |
|---|---|---|---|---|---|---|---|---|
| | den DF | F | Pr | estim. | den DF | F | Pr | estim. |
| *Fixed effects* | | | | | | | | |
| Crown cover | 130 | 6.53 | * | (-)0.15 | 173 | 9.11 | ** | (-)0.14 |
| Slope | 151 | 1.23 | ns | 0.06 | 168 | 2.25 | ns | (-)0.06 |
| Surface cover | | | | | | | | |
| - Biocrust | 151 | 50.2 | *** | (-)0.38 | 159 | 8.11 | ** | (-)0.12 |
| - Stone | 136 | 10.3 | ** | (-)0.19 | 188 | 1.66 | ns | (-)0.06 |
| SOM | 44 | 5.71 | * | (-)0.08 | 72 | 2.43 | ns | 0.12 |
| Rainfall | 95 | 5.46 | * | (-)0.08 | 302 | 13.2 | *** | 0.14 |
| Tree species richness | 22 | 0.46 | ns | 0.05 | 68 | 0.11 | ns | (-)0.03 |
| | | | | | *SD* | | | |
| *Random effects* | *SD* | | *Variance* | | | | *Variance* | |
| Precip. event : plot | 0.199 | | 0.040 | | 0.537 | | 0.288 | |
| Tree composition | 0.292 | | 0.085 | | 0.000 | | 0.000 | |
| Site | 0.466 | | 0.217 | | 0.443 | | 0.196 | |
| Plot : ROP | 0.441 | | 0.195 | | 0.269 | | 0.073 | |

SOM: soil organic matter; denDF: denominator degrees of freedom; F: F value; Pr: significance; estim.: estimates













**Figures**

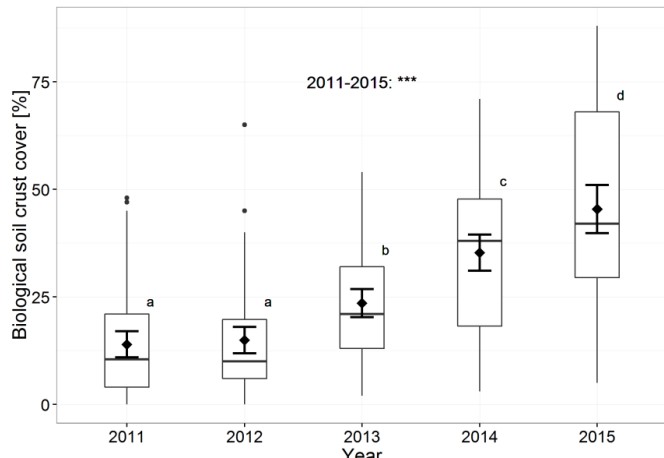


**Figure 1: The development of biological soil crust cover in runoff plots of the BEF China experiment from 2011 to**
**2015 in Xingangshan, Jiangxi Province, PR China (n=350). Horizontal lines within boxplot represent medians and**
**diamonds represent means with standard error bars. Points signify outliers and small letters significant differences**
**(p<0.001).**









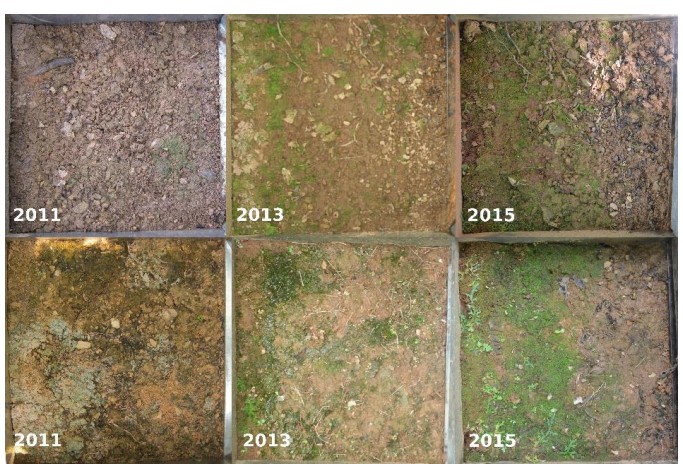


**Figure 2: Successional stages of biological soil crusts in two exemplary runoff plots (top row and bottom row, 0.4 m ×**

**0.4 m each) in 2011, 2013 and 2015 (from left to right) at the BEF China experiment in Xingangshan, Jiangxi**

**Province, PR China.**












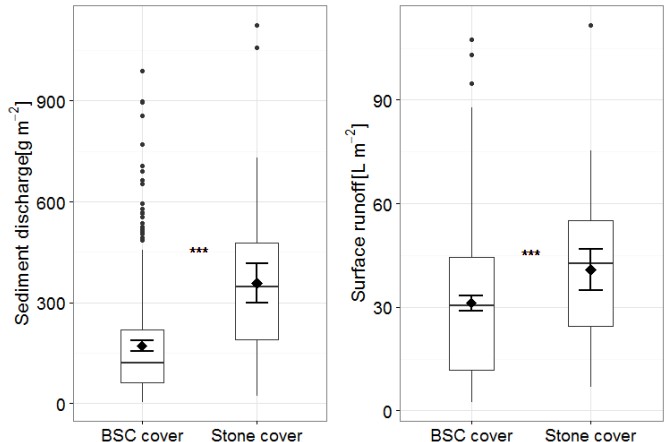


**Figure 3: The influence of runoff plots dominated by biological soil crust cover (BSC, n=281) and stone cover (n=53)**
**on sediment discharge and surface runoff in Xingangshan, Jiangxi Province, PR China in 2013. Horizontal lines**
**within box plots represent median and diamonds represent mean with standard error bars.**