# Peer review of "Bryophyte-dominated biological soil crusts mitigate soil erosion"

_Biogeosciences, 2017_

## Referee Comment (RC1) · Anonymous Referee #1 · 26 Apr 2017

Compared with lots of reports regarding biocrusts from drylands, little topics were discussed in humid region. This paper selected a novel issue and conducted a scientific experiment, and whereby some clear but not unexpected results were concluded. My suggestions are as follows to improve the manuscript:

1.while authors analyzed the influence of vegetation, soil, and terrain on biocrusts cover, they should emphasized the interactive impact of different factors, but not only single factor's influence;

2.the authors must quatitatively describe the effects of biocrusts on sediment and runoff, and also in the discussion section, they should compared their results with other researchers' conclusions;

3.generally, this manuscript present too much emphasis on qualitative description instead of quatitative analysis;

4.the experiment was conducted in PR China, however, there is none of authors or affiliation from Chinese territory, which is not logical or even not permitted by China goverment;

I suggest that authors improve the paper according to the comments above at least.

---

## Referee Comment (RC2) · Anonymous Referee #2 · 27 Apr 2017

I have received your manuscript "Bryophyte-dominated biological soil crusts mitigate soil erosion in an early successional Chinese subtropical forest" for review. There are elements to like about this paper, especially the fact that this is one of the rare studies that addresses the influence and role of biological soil crusts on in early secondary succession after severe human impact. Before the manuscript can be considered for publication in Biogeosciences, the authors should nevertheless consider some general comments and rework parts of the manuscript. There is a general question as to how the soil crust in this studied can be referred to. In the introduction (first sentence) the authors refer to the importance of biocrusts in many ecosystems. By checking the reference and also other major biocrust research and literature it becomes obvious that

the biocrust in this study is very special because it occurs in a forest. Such ecosystems are dominated by trees are their sheer occurrence indicates a high water availability. So, here it comes to a contradiction to the definition of a biocrust, which were recently defined to occur "in regions where water availability limits vascular plant cover" (Weber et al. 2016 - same reference as used by the authors). Almost every other biocrust study is taking place in drylands or at least areas where an arid element occurs. In the most recent review about biocrust distribution patterns by Bowker et al. 2016, the presence of biocrust is discussed as a function of effective precipitation within semiarid, arid and hyperarid ecosystems, certainly not in the humid forest with a mean precipitation of 1635 mm as in this case. This makes this study very special and requires that the authors explain very precisely why they discuss the topic in the biocrust background. It should be stated in the manuscript that biocrusts only contribute a minor ecological role in this ecosystem, certainly because of their low biomass and soil penetration depth, compared with trees and it should be taken into account that the trees are the major driver of this ecosystem. The presence of the mosses and algae may certainly have an effect as shown in this paper, but this should be seen in the bigger context and appropriately assigned. In the ongoing introduction, the authors explicitly describe the role of biocrusts in early succession, while the study site cannot be referred to as in early succession because of the existence of trees, that indicate quite a late successional stage. Nevertheless, the study site is special, because the trees were artificially planted, so the soil itself remains at an early successional stage (straight after disturbance) while the occurring vascular vegetation is at a later stage, due to human impact (at least the trees. What other vegetation occurs?). A more detailed explanation of the hypothetical background should be taken into consideration. Additionally, it should be clearly defined which ecological process is in the focus of this study. From the study background, the most reasonable is secondary succession after human disturbance. Within this, the biocrust may occur as one of the initial players, thus it will provide the basis for other plants but also disappear with ongoing succession. This should be stated. The hypotheses should come with explanations or at least theoretical background. In Hyp

1 is the parameter tree growth or canopy cover or light intensity? Hyp 2, what is the exact expectation here? This sound very vague. Please be precise. In general, the introduction could benefit from more precise statements and direct explanations. In the moment many sections ready like overall summaries rather than leading to explicit research questions for the study. Material and Methods 135: which were the determinants for the crust types Results 210: Please explain why the existence of vascular plants indicates any developmental stage of biocrust? Is this climax or are the plants taking over and the crust will disappear? If this is the case you should refer to the developmental stage of the vegetation in general and the crusts occur only for a little while. Discussion Overall the discussion could clearly benefit from more explicit arguments of the given results rather than summarising literature. Sections read like reviews and summaries of recent literature. Could you please discuss your own data and indicate what information these add to the existing knowledge? Here I strongly agree with the first reviewer how also stated that the discussion needs improvement. Please explain more detailed what your own findings mean and implement. Additionally, you might want to consider reading more about bryophytes and their growth in forest understories. Surely the discussion could benefit from some comparisons for growth rates and microenvironments. For the implication of the story, the authors should clearly underline, that this is a case study in a single, very special subtropical forest ecosystem and therefore findings cannot easily be extrapolated to other systems. In the first section of the discussion, statements are made, that do not refer to the presented data. 243: biocrusts were highly competitive 244: biocrusts prepared the upper soil layer 246: tree growth provides shade and protection from the wind, which then leads to advancement in biocrust development (Please explain that you use crown cover as a proxy for shading. Are there data about the wind?). Other 260: Replace competitive by coexisting. 261-265: The authors provide a very nice list of moss species for this study and assume here that species composition changes with decreasing light availability. It would be excellent to underline this finding by data. Can you provide data that show this shift of species? The statement could significantly gain importance if the change

in species could be correlated to the decreasing light. As it stands now, it reads like an assumption. 279-280: Irrelevant for the study. Can be deleted. Additional minor comments: Some of the writing does not seem to be appropriate. Please reconsider 34: "Our" experimental forest ecosystem 36: Biocrust "covers" were still increasing 42: quickly colonise gaps in" higher vegetation layers" – what do you mean with layers? Which gaps are closed? 207: "traces of lichens" 210: organisms were found in minor numbers – is this fewer species, individuals or coverage? 243: early stage of the ecosystem 283: fasten themselves on the soil surface 329: They developed quickly to later-stages

---

## Referee Comment (RC3) · Anonymous Referee #3 · 2 May 2017

The topic of this paper is likely to have a substantial impact. However, it is very difficult to follow the numbers of sample replicates across the study. Perhaps a table would help where the hierarchy of sampling is broken down and all in one place. The authors also put a lot of influence on canopy cover dictating cover of biocrusts but their abiotic variables are likely influencing the canopy cover. These things should be addressed together or the abiotic setting should be controlled for when looking at canopy cover. The disturbance that is mentioned, that is the reason for the development of biocrusts is not described.

Specific comments: Line 114: The hypotheses could be stated more clearly. (1) "Biocrusts ARE widely developED (2) "The development of biocrust is influenced by

BOTH the surrounding vegetation cover AND THE soil and terrain attributes." Line 173- It is not clear is the analyses met the assumptions of ANOVA. Line 185- "Than" should be "then". Line 239- I thought that this paper was primarily about soil erosion and biocrusts but that it not clear from the first paragraph of the discussion. Line 240- It is confusing to state hypotheses by numbers but quickly paraphrasing the hypothesis would make interpretation easier for the reader. Line 241- Is there any pre-disturbance data? It is hard to understand the connection between interspaces and disturbance without some description of the pre-disturbance structure of the vegetation. Line 271- Cite Condon and Pyke 2016, who have been able to restore a great deal of moss cover very quickly. Line 321- You would have a stronger close if you finished with the sentence that ends here. It's also unclear given your findings if there is much of a need to restore biocrusts since you saw recovery of bryophytes really quickly. Line 335- You should remind the reader here of your scale as this likely influenced the effects of soil attributes. The authors need to work on the storyline of the paper as well.

---

## Author Comment (AC1) · 17 May 2017

Thank you very much for taking the time to revise this manuscript and for giving this positive evaluation with constructive comments.

In this reply, we address your four general suggestions:

1. "While authors analyzed the influence of vegetation, soil, and terrain on biocrusts cover, they should emphasized the interactive impact of different factors, but not only single factor's influence"

- This work benefits from the large dataset of the BEF China project and we tried to include a high number of single influences. Nevertheless, we agree with you that

interactions of different factors are of high importance and more explanations regarding interactions are needed in the manuscript. We already tried to address this issue by using linear mixed effects models, which account for a combination of fixed and random effects. We will now widen the discussion part of the manuscript in this regard and try to better explain combined effects of single factors.

2. "the authors must quatitatively describe the effects of biocrusts on sediment and runoff, and also in the discussion section, they should compared their results with other researchers' conclusions"

- We agree with you and reviewer #2. More quantitative description will be added to the manuscript as well as further comparisons with other studies in particular. Nevertheless, we want to underline, that, as you and both other reviewers stated, there are only very few studies on biocrust development under forest in this climate to compare with.

3. "generally, this manuscript present too much emphasis on qualitative description instead of quatitative analysis"

- We will consider this point for a revised version of the manuscript and try to give this work a better-balanced ratio of quantitative and qualitative aspects (see comment above).

4. "the experiment was conducted in PR China, however, there is none of authors or affiliation from Chinese territory, which is not logical or even not permitted by China goverment"

- The BEF China project is a joint Sino-German research project with a high level of close cooperation between Chinese and German universities. Thus, our author's team also includes a Chinese national. Nevertheless, we agree that affiliations are not sufficiently presented and we are currently in contact with our Chinese partners for this purpose.

Thus, we generally agree with your comments and we will carefully consider all your suggestions to improve the manuscript.

---

## Author Comment (AC2) · 18 May 2017

Thank you for your positive evaluation and helpful comments on our manuscript, which we want to address one by one:

"The topic of this paper is likely to have a substantial impact."

- Thank you for this assessment.

"However, it is very difficult to follow the numbers of sample replicates across the study. Perhaps a table would help where the hierarchy of sampling is broken down and all in one place."

- We agree that sample numbers appear somewhat confusing, which is because the study was initially not designed to investigate BSC development. We will add a table to clarify sample numbers in the revised version of the manuscript.

"The authors also put a lot of influence on canopy cover dictating cover of biocrusts but their abiotic variables are likely influencing the canopy cover. These things should be addressed together or the abiotic setting should be controlled for when looking at canopy cover."

- Abiotic factors were mainly used to explain biocrust covers. We agree that they are likely influencing tree development too and their influence on canopy cover is not sufficiently addressed. We will widen the discussion in this regard.

"The disturbance that is mentioned, that is the reason for the development of biocrusts is not described."

- The potential natural vegetation of this region is a subtropical broadleaved forest with dominating evergreen species, which was replaced by a commercial forest plantation in the 1980's. This plantation was then clear-cut in 2008 and the BEF-China experiment was established (see methods 2.1, Bruelheide et al. 2014, Yang et al. 2013). We will present more information on the vegetation disturbance on the study site in the revised manuscript (see below).

Specific comments:

"Line 114: The hypotheses could be stated more clearly. (1) "Biocrusts ARE widely developED (2) "The development of biocrust is influenced by BOTH the surrounding vegetation cover AND THE soil and terrain attributes." "

- Hypotheses will be stated more clearly, according to the reviewer's suggestion.

"Line 173- It is not clear is the analyses met the assumptions of ANOVA."

- The dataset was tested and met the assumptions for ANOVA. We will add further

information about normality and multi-collinearity to this paragraph.

"Line 185- "Than" should be "then"."

- Changed

"Line 239- I thought that this paper was primarily about soil erosion and biocrusts but that it not clear from the first paragraph of the discussion."

- The paper is structured along the three hypotheses "(1) BSC extend under forest", "(2) Influence of vegetation, soil and terrain on BSC development", "(3) Impact on soil erosion". Thus, we used the same order in the results and discussion part. The title is pointing more on soil erosion, though, as this was the most important finding in our opinion. Nevertheless, BSC development and e.g. classified moss species are of high importance, too.

"Line 240- It is confusing to state hypotheses by numbers but quickly paraphrasing the hypothesis would make interpretation easier for the reader."

- We agree that paraphrasing the hypotheses in the discussion part would enhance the comprehensibility for the reader. Thus, we will add short repetitions of the hypotheses at the beginning of each discussion paragraph.

"Line 241- Is there any pre-disturbance data? It is hard to understand the connection between interspaces and disturbance without some description of the pre-disturbance structure of the vegetation."

- See above. Unfortunately, we do not have any data on BSCs and BSC development derived before the establishment of the experiment in 2009-2010.

"Line 271- Cite Condon and Pyke 2016, who have been able to restore a great deal of moss cover very quickly."

- Thank you for pointing out further literature. The work of Condon and Pyke gives very valuable insights on moss development after vegetation disturbance and fits well in our

context. We will consider this study for the revised manuscript.

"Line 321- You would have a stronger close if you finished with the sentence that ends here. It's also unclear given your findings if there is much of a need to restore biocrusts since you saw recovery of bryophytes really quickly."

- That is true for our specific study and this subtropical forest environment. Nevertheless, biocrust restoration could be applied in other forest environments to protect bare soils e.g. in skid trails after timber harvesting. Thus, this might be interesting to look at in further studies and other climates. Nevertheless, we will revise the phrasing of the last two sentences.

"Line 335- You should remind the reader here of your scale as this likely influenced the effects of soil attributes. The authors need to work on the storyline of the paper as well."

- We will add further information about scale and try to improve the manuscript as the reviewer suggests.

---

## Author Comment (AC3) · 24 May 2017

Many thanks for your critical and constructive review. We will consider your comments and rework the manuscript as suggested. Please find more detailed responses to your remarks below:

1.) There is a general question as to how the soil crust in this studied can be referred to. In the introduction (first sentence) the authors refer to the importance of biocrusts in many ecosystems. By checking the reference and also other major biocrust research and literature it becomes obvious that the biocrust in this study is very special because it occurs in a forest. Such ecosystems are dominated by trees are their sheer occurrence indicates a high water availability. So, here it comes to a contradiction to the

[Figure]

definition of a biocrust, which were recently defined to occur "in regions where water availability limits vascular plant cover" (Weber et al. 2016 - same reference as used by the authors). Almost every other biocrust study is taking place in drylands or at least areas where an arid element occurs. In the most recent review about biocrust distribution patterns by Bowker et al. 2016, the presence of biocrust is discussed as a function of effective precipitation within semiarid, arid and hyperarid ecosystems, certainly not in the humid forest with a mean precipitation of 1635 mm as in this case. This makes this study very special and requires that the authors explain very precisely why they discuss the topic in the biocrust background.

- We would like to thank the reviewer for his/her overall positive review and especially for the comment above. It indeed is right that the biocrust described in this manuscript is a special one, as it occurs within a forest. Biological soil crusts are defined to occur in arid environments and in places where arid (microclimatic) conditions are met. Thus, biocrusts are mostly described for dryland regions, where they form a key and enduring ecosystem component. In addition to this main habitat, however, biocrusts have also been described as a transient feature in mesic environments, where annual rainfall amounts are larger, but where biocrusts may form after major singular or repeated disturbance events. Example for such transient biocrusts are e.g. these of dry grasslands ("Mainfränkische Trockenrasen", Germany) and biocrusts occurring on Öland, Sweden (Büdel et al., 2014), but also biocrusts in former mining areas (Fischer et al., 2014). In all these habitats, biocrusts developed due to severe disturbance caused by trampling and driving, continuous grazing, soil removal and biomass removal. Once these regular disturbance events stop, a dense layer of vascular vegetation would develop and replace the biocrusts. In the current study, an existing forest has been clear-cut and trees have been replanted. After this severe disturbance, transient biocrusts developed, which could grow on the bare ground under fairly sunny conditions. However, we already observed an alteration in species inventory and expect, that eventually this biocrust will disappear again, being replaced by vascular vegetation (in light forests) or buried under persisting leaf litter (under darker conditions).

2.) It should be stated in the manuscript that biocrusts only contribute a minor ecological role in this ecosystem, certainly because of their low biomass and soil penetration depth, compared with trees and it should be taken into account that the trees are the major driver of this ecosystem. The presence of the mosses and algae may certainly have an effect as shown in this paper, but this should be seen in the bigger context and appropriately assigned.

- We fully agree with the comment, that biomass of biocrusts within this habitat is clearly negligible. Nevertheless, we observed that biocrusts play a key role within this early-successional system. By minimizing erosion and stabilizing the soil surface, they effectively preserve soil fertility within this disturbed forest habitat. Thus, despite their low biomass, we consider biocrusts to fulfil key ecosystem services within this habitat.

3.) In the ongoing introduction, the authors explicitly describe the role of biocrusts in early succession, while the study site cannot be referred to as in early succession because of the existence of trees, that indicate quite a late successional stage. Nevertheless, the study site is special, because the trees were artificially planted, so the soil itself remains at an early successional stage (straight after disturbance) while the occurring vascular vegetation is at a later stage, due to human impact (at least the trees. What other vegetation occurs?). A more detailed explanation of the hypothetical background should be taken into consideration.

- We agree that the special circumstances of the study site indeed need to be described in more detail in the paper. From a forestry perspective, the study site was referred to as an early successional tree plantation in 2013 (see Bruelheide et al. 2014). To establish the experiment and help the tree saplings ($\sim$5-15 cm) grow in their first years, shrubs and coppices have been weeded from 2010 to 2012. Thus, it is an artificial setup, but following common practice in forestry and tree plantations of this area. We will give a more detailed explanation in the revised manuscript.

4.) Additionally, it should be clearly defined which ecological process is in the focus
of this study. From the study background, the most reasonable is secondary succession after human disturbance. Within this, the biocrust may occur as one of the initial players, thus it will provide the basis for other plants but also disappear with ongoing succession. This should be stated. The hypotheses should come with explanations or at least theoretical background. In Hyp 1 is the parameter tree growth or canopy cover or light intensity? Hyp 2, what is the exact expectation here? This sound very vague. Please be precise.

- We agree and thank you again for this very helpful and to-the-point comment. We do expect that biocrusts are an initial player and they will disappear with ongoing succession (see e.g. line 267). Actually, we were surprised that their extend was still increasing after five years of tree growth as some tree species are already up to 7 m height. Nevertheless, we will clarify the process in focus and sharpen our hypotheses.

5.) In general, the introduction could benefit from more precise statements and direct explanations. In the moment, many sections ready like overall summaries rather than leading to explicit research questions for the study. Material and Methods 135: which were the determinants for the crust types

- Crust types were classified in the field based on appearance, functional groups and species composition, if possible. Further details will be added to the manuscript.

6.) Results 210: Please explain why the existence of vascular plants indicates any developmental stage of biocrust? Is this climax or are the plants taking over and the crust will disappear? If this is the case you should refer to the developmental stage of the vegetation in general and the crusts occur only for a little while.

- We believe that vascular plants like Selaginella and Poaceae will take over and biocrust covers will decline, even without a continuous leaf litter layer. We will add further information (see above).

7.) Discussion Overall the discussion could clearly benefit from more explicit arguments of the given results rather than summarising literature. Sections read like reviews and summaries of recent literature. Could you please discuss your own data and indicate what information these add to the existing knowledge? Here I strongly agree with the first reviewer how also stated that the discussion needs improvement. Please explain more detailed what your own findings mean and implement.

- We agree with you and reviewer #1 and we will sharpen the discussion and strengthen the focus on our own data. More quantitative description will be added to the manuscript (see above).

8.) Additionally, you might want to consider reading more about bryophytes and their growth in forest understories. Surely the discussion could benefit from some comparisons for growth rates and microenvironments. For the implication of the story, the authors should clearly underline, that this is a case study in a single, very special subtropical forest ecosystem and therefore findings cannot easily be extrapolated to other systems.

- We agree that findings cannot be extrapolated to other ecosystems and it is not our objective to do so. Nevertheless, we believe that they could be compared to other forest plantations in subtropical China and it would be very interesting to evaluate the BSC occurrence in nearby forests right after clear-cut and replantation. Furthermore, we will add more literature about bryophyte development in forests. We fully agree that this study could benefit from comparisons of growth rates, for example. Nevertheless, there are not so much other studies on BSCs in those subtropical climates, to our knowledge.

9.) In the first section of the discussion, statements are made, that do not refer to the presented data. 243: biocrusts were highly competitive 244: biocrusts prepared the upper soil layer 246: tree growth provides shade and protection from the wind, which then leads to advancement in biocrust development (Please explain that you use crown cover as a proxy for shading. Are there data about the wind?). Other 260:

Replace competitive by coexisting. 261-265: The authors provide a very nice list of moss species for this study and assume here that species composition changes with decreasing light availability. It would be excellent to underline this finding by data. Can you provide data that show this shift of species? The statement could significantly gain importance if the change in species could be correlated to the decreasing light. As it stands now, it reads like an assumption.

- We will add further data and better explain the already presented data. We agree that adding more data about single species from e.g. 2011 and 2015 would be very useful and beneficial. Unfortunately, we do not have any further detailed species lists as it was not possible to sample biocrusts but in 2013. Classifications in other years were done by visual judgement in the field.

10.) 279-280: Irrelevant for the study. Can be deleted.

- Agreed. Will be deleted.

11.) Additional minor comments: Some of the writing does not seem to be appropriate. Please reconsider 34: "Our" experimental forest ecosystem 36: Biocrust "covers" were still increasing 42: quickly colonise gaps in" higher vegetation layers" – what do you mean with layers? Which gaps are closed? 207: "traces of lichens" 210: organisms were found in minor numbers – is this fewer species, individuals or coverage? 243: early stage of the ecosystem 283: fasten themselves on the soil surface 329: They developed quickly to later-stages

- Thank you for pointing out mistakes and incomprehensibilities in writing. We will change the writing according to your comments and the whole manuscript will be counter-checked by an English native speaker after revision.

---

## Author Response (AR1)

**Response to reviewer comments on manuscript bg-2017-99: "Bryophyte-dominated biological soil**
**crusts mitigate soil erosion in an early successional Chinese subtropical forest"**

We would like to thank the reviewers for their helpful comments, which greatly improved our
manuscript. We have prepared a revised manuscript where we account for all points raised by the
reviewers, as described below. We show the reviewers' comments in grey text, while our responses are
formatted as standard text. Line indications refer to the revised manuscript without marked changes.

Anonymous referee #1:

Thank you very much for taking the time to revise this manuscript and for giving this positive evaluation
with constructive comments.
Compared with lots of reports regarding biocrusts from drylands, little topics were discussed in humid
region. This paper selected a novel issue and conducted a scientific experiment, and whereby some clear
but not unexpected results were concluded. My suggestions are as follows to improve the manuscript:
1.  while authors analyzed the influence of vegetation, soil, and terrain on biocrusts cover, they should
emphasized the interactive impact of different factors, but not only single factor's influence;
-   This work benefits from the large dataset of the BEF China project and we tried to include a
high number of single influences. Nevertheless, we agree with you that interactions of
different factors are of high importance and more explanations regarding interactions are
needed in the manuscript. We already tried to address this issue by using linear mixed
effects models, which account for a combination of fixed and random effects (lines 182-187,
188-196, 223-226, Table 3, 233-238, Table 4). We now widened the discussion part of the
manuscript in this regard and tried to better explain combined effects of single factors, such
as vegetation (crown cover and LAI) or terrain (e.g. northness) (lines 286-290, 299-310, 311-
318).
2.  the authors must quatitatively describe the effects of biocrusts on sediment and
runoff, and also in the discussion section, they should compared their results with other
researchers' conclusions;
-   Thank you very much for this legitimate comment. We widened the description of our
results, demonstrating the effects of biocrusts on soil erosion, by presenting measured
values of runoff and sediment delivery (see lines 231-233). We also widened the discussion
section of the manuscript by including additional comparisons with other studies (lines 320-
324, 339-343, 344-348).
Nevertheless, as correctly stated by all reviewers, these comparisons have to remain
incomplete, as there are only very few studies on biocrust development in forests under
mesic conditions and their impact on soil erosion.

3. generally, this manuscript present too much emphasis on qualitative description instead of
quatitative analysis;
- We considered this aspect in the revised version of the manuscript and indeed put a larger
    emphasis on our quantitative results and widened the discussion (see above).

4. the experiment was conducted in PR China, however, there is none of authors or affiliation from
Chinese territory, which is not logical or even not permitted by China government
- The BEF China project is a joint Sino-German research project with a high level of close
    cooperation between Chinese and German universities. Thus, our authors' team already
    included a Chinese national, who was originally based at the Institute of Soil Science, CAS,
    Nanjing. Nevertheless, we agree that affiliations were not sufficiently presented and our
    Chinese cooperation partner and supervisor of fieldworks in China Prof. Xuezheng Shi from
    the Institute of Soil Science in Nanjing agreed to join the authors' team and contributed to
    the revision of the manuscript (see affiliations).

Anonymous referee #2:

Many thanks for your critical and constructive review. We considered your comments and reworked the
manuscript as suggested. Please find more detailed responses to your remarks below.

1.) I have received your manuscript "Bryophyte-dominated biological soil crusts mitigate soil erosion in
an early successional Chinese subtropical forest" for review. There are elements to like about this paper,
especially the fact that this is one of the rare studies that addresses the influence and role of biological
soil crusts on in early secondary succession after severe human impact. Before the manuscript can be
considered for publication in Biogeosciences, the authors should nevertheless consider some general
comments and rework parts of the manuscript. There is a general question as to how the soil crust in
this studied can be referred to. In the introduction (first sentence) the authors refer to the importance
of biocrusts in many ecosystems. By checking the reference and also other major biocrust research and
literature it becomes obvious that the biocrust in this study is very special because it occurs in a forest.
Such ecosystems are dominated by trees are their sheer occurrence indicates a high water availability.
So, here it comes to a contradiction to the definition of a biocrust, which were recently defined to occur
"in regions where water availability limits vascular plant cover" (Weber et al. 2016 - same reference as
used by the authors). Almost every other biocrust study is taking place in drylands or at least areas
where an arid element occurs. In the most recent review about biocrust distribution patterns by Bowker
et al. 2016, the presence of biocrust is discussed as a function of effective precipitation within semiarid,
arid and hyperarid ecosystems, certainly not in the humid forest with a mean precipitation of 1635 mm
as in this case. This makes this study very special and requires that the authors explain very precisely
why they discuss the topic in the biocrust background.

-     We would like to thank the reviewer for this overall positive review and especially for the
 comment above. It indeed is right that the biocrust described in this manuscript is a special one,
 as it occurs within a forest. Biological soil crusts are defined to occur in arid environments and in
 places where arid (microclimatic) conditions are met. Thus, biocrusts are mostly described for
 dryland regions, where they form a key and enduring ecosystem component. In addition to this
 main habitat, however, biocrusts have also been described as a transient feature in mesic
 environments, where annual rainfall amounts are larger, but where biocrusts may form after
 major singular or repeated disturbance events. Examples for such transient biocrusts are e.g.
 these of dry grasslands ("Mainfränkische Trockenrasen", Germany) and biocrusts occurring on
 Öland, Sweden (Büdel et al., 2014), but also biocrusts in former mining areas (Fischer et al.,
 2014). In all these habitats, biocrusts developed due to severe disturbance caused by trampling
 and driving, continuous grazing, soil removal and biomass removal. Once these regular
 disturbance events stop, a dense layer of vascular vegetation would develop and replace the
 biocrusts.
 In the current study, an existing forest has been clear-cut and trees have been replanted. After
 this severe disturbance, transient biocrusts developed, which could grow on the bare ground
 under fairly sunny conditions. However, we already observed an alteration in species inventory
 and expect, that eventually these biocrusts will disappear again, being replaced by vascular
 vegetation (in light forests) or buried under persisting leaf litter (under darker conditions) (lines
269-276).
In the revised version, we sharpened the discussion and conclusion about biocrusts in mesic
environments and explained the relationship between biocrust occurrence and vegetation
disturbances (lines 64-67, 69-76, 81-83, 249-252, 269-276, 361-364).

2.) It should be stated in the manuscript that biocrusts only contribute a minor ecological role in this
ecosystem, certainly because of their low biomass and soil penetration depth, compared with trees and
it should be taken into account that the trees are the major driver of this ecosystem. The presence of
the mosses and algae may certainly have an effect as shown in this paper, but this should be seen in the
bigger context and appropriately assigned.

-     We fully agree with the comment, that biomass of biocrusts within this habitat is nearly
negligible. Nevertheless, we observed that biocrusts play a key role within this early-
successional system. By minimizing erosion and stabilizing the soil surface, they effectively
preserve soil fertility within this disturbed forest habitat. Thus, despite their low biomass, we
consider biocrusts to fulfil key ecosystem services within this habitat. Nevertheless, we agree
that the bigger context should be better assigned and we added further information on that
issue to the discussion (lines 324-326, 341-343).

3.) In the ongoing introduction, the authors explicitly describe the role of biocrusts in early succession,
while the study site cannot be referred to as in early succession because of the existence of trees, that
indicate quite a late successional stage. Nevertheless, the study site is special, because the trees were artificially planted, so the soil itself remains at an early successional stage (straight after disturbance)
while the occurring vascular vegetation is at a later stage, due to human impact (at least the trees. What
other vegetation occurs?). A more detailed explanation of the hypothetical background should be taken
into consideration.

-    We agree that the special circumstances of the study site indeed need to be described in more
detail in the paper. From a forestry perspective, the study site was referred to as an early
successional tree plantation in 2013 (see Bruelheide et al. 2014). To establish the experiment
and help the tree saplings (~5-15 cm height) grow in their first two years, shrubs and coppices
have been weeded from 2010 to 2012. Thus, it is an artificial setup, but following common
practice in forestry and tree plantations of this area. Understory growth was generally low, with
smaller patches of e.g. *Miscanthus* at southward slopes and only little leaf abscission from trees
in the first years. We gave a more detailed explanation in the revised manuscript (lines 131-137,
146-147).

4.) Additionally, it should be clearly defined which ecological process is in the focus of this study. From
the study background, the most reasonable is secondary succession after human disturbance. Within
this, the biocrust may occur as one of the initial players, thus it will provide the basis for other plants but
also disappear with ongoing succession. This should be stated.

-    We agree and thank you again for this very helpful comment. Some further statements were
added (line 110 f). We do expect that biocrusts are an initial player (lines 247-249, 253-255, 255-
258) and they will disappear with ongoing succession (line 269-270, 273-276). Actually, we were
surprised that their extent was still increasing after five years of tree growth, when some trees
were already up to a height of 7 m.

5.) The hypotheses should come with explanations or at least theoretical background. In Hyp 1 is the
parameter tree growth or canopy cover or light intensity? Hyp 2, what is the exact expectation here?
This sound very vague. Please be precise.

-    We adapted the introduction to give more theoretical background (lines 57ff) and sharpened
the hypotheses (116-121). Regarding Hyp 2, we tried to focus on the humid forest environment
and to keep it somewhat more general, as we will not be able to explain all influences based on
our data set. The pH for example, influences crust components (lichens, algae etc.) differently
and e.g. different bryophyte species show variable responses on pH changes.

6.) In general, the introduction could benefit from more precise statements and direct explanations. In
the moment, many sections ready like overall summaries rather than leading to explicit research
questions for the study.

-     Thank you for this comment. We adapted the introduction and tried to be more precise and to
  better link the introductory part to the statement of hypotheses. We also included assumptions
  to each paragraph, leading to the hypothesis generation (lines 81-83, 88-91, 107-109).

7.)

Material and Methods 135: which were the determinants for the crust types

-     Biocrusts were described in the field based on appearance, functional groups and species
  composition and biocrust types determined based on the dominating autotrophic component.
  Further details were added to the manuscript (lines 140-142).

Results 210: Please explain why the existence of vascular plants indicates any developmental stage of
biocrust? Is this climax or are the plants taking over and the crust will disappear? If this is the case you
should refer to the developmental stage of the vegetation in general and the crusts occur only for a little
while.

-     The existence of vascular plants does not indicate a developmental stage of biocrusts; the
  expression was somewhat misleading. Instead, we believe that vascular plants like *Selaginella*
  and *Poaceae* will take over and biocrust cover will decline, even without a continuous leaf litter
  layer (see above). We changed the phrasing accordingly (lines 215 - 216)

Discussion Overall the discussion could clearly benefit from more explicit arguments of the given results
rather than summarising literature. Sections read like reviews and summaries of recent literature. Could
you please discuss your own data and indicate what information these add to the existing knowledge?
Here I strongly agree with the first reviewer how also stated that the discussion needs improvement.
Please explain more detailed what your own findings mean and implement.

-     We agree with you and reviewer #1 (see comment above) and sharpened the discussion and
  strengthened the focus on our own data. Major parts of the discussion have been rearranged
  and widened (e.g. lines 249-251, 253-255, 256-258, 269-272, 274-276, 277-278, 290-294, 302-
  304, 309-310, 311-313, 316-318, 321-324, 324-326, 335-339, 341-343, 350-351).

Additionally, you might want to consider reading more about bryophytes and their growth in forest
understories. Surely the discussion could benefit from some comparisons for growth rates and
microenvironments. For the implication of the story, the authors should clearly underline, that this is a
case study in a single, very special subtropical forest ecosystem and therefore findings cannot easily be
extrapolated to other systems.

-    We agree with you that findings cannot be extrapolated to other ecosystems and it is not our
objective to do so. We clarified this point in the revised conclusion (lines 361-363). Nevertheless,
we believe that they could be compared to other forest plantations in subtropical China and it
would be very interesting to evaluate the biocrust occurrence in other forest ecosystems after
clear-cut and replantation of trees.
Furthermore, we added more literature about biocrust recovery (Read et al 2011, Read et al
2016, Eldridge 1998) and bryophyte development in forests (Fenton and Frego 2005, Goffinet
and Shaw 2009, Spitale 2017, Tinya and Odor 2016, Gilliam 2007). We fully agree that this study
could benefit from comparisons of growth rates, for example. Nevertheless, there are not so
many other studies on biocrust recovery in those subtropical climates and to our knowledge,
generally studies in forest environments are rare (except work done by Read et al. and Eldridge
from Australian semi-arid woodlands, see above).

In the first section of the discussion, statements are made, that do not refer to the presented data. 243:
biocrusts were highly competitive 244: biocrusts prepared the upper soil layer 246: tree growth provides
shade and protection from the wind, which then leads to advancement in biocrust development (Please
explain that you use crown cover as a proxy for shading. Are there data about the wind?). Other 260:
Replace competitive by coexisting. 261-265: The authors provide a very nice list of moss species for this
study and assume here that species composition changes with decreasing light availability. It would be
excellent to underline this finding by data. Can you provide data that show this shift of species? The
statement could significantly gain importance if the change in species could be correlated to the
decreasing light. As it stands now, it reads like an assumption.

-    We changed the discussion according to your suggestions (lines 246-263). Crown cover is a
proxy for shading and there is no detailed data about wind speed or direction.
Furthermore, we agree that adding more data about single species from e.g. 2011 and 2015
would be very useful and beneficial. Unfortunately, we do not have any further detailed species
lists as it was not possible to sample biocrusts but in 2013. Classifications of biocrust types in
other years were done by visual judgement in the field. Thus, we cannot substantiate the
species shift with further data, but would like to do so in further studies.

279-280: Irrelevant for the study. Can be deleted.

-    Deleted.

Additional minor comments:

Some of the writing does not seem to be appropriate. Please reconsider

34: "Our" experimental forest ecosystem

36: Biocrust "covers" were still increasing

42: quickly colonise gaps in" higher vegetation layers" – what do you mean with layers? Which gaps are
closed?

207: "traces of lichens"

210: organisms were found in minor numbers – is this fewer species, individuals or coverage?

243: early stage of the ecosystem

283: fasten themselves on the soil surface

329: They developed quickly to later-stages

-   Thank you for pointing out mistakes and incomprehensibilities in writing. We changed the
writing according to your comments and counterchecked the manuscript.
-   42: Higher vegetation layers in this context are the tree or shrub layer and gaps occur by treefall
or tree cutting. 220: This is fewer species (then bryophyte species).

Anonymous referee #3:

Thank you very much for your review and the very valuable suggestions to improve the manuscript.

1.) The topic of this paper is likely to have a substantial impact. However, it is very difficult to follow the
numbers of sample replicates across the study. Perhaps a table would help where the hierarchy of
sampling is broken down and all in one place.

-   Thank you for this comment. We agree that sample numbers may appear somewhat confusing,
which is due to the fact that the study was initially not designed to investigate biocrust
development. Within the BEF China experiment, 34 research plots were used and equipped with
five runoff plots each (thus 170 ROPs in total). Measuring four rainfall events in 2013, we came
up with an n=334 of valid measurements for the erosion assessment. From 2011 to 2015, a
selection of 70 ROPs could be continuously checked for biocrusts. We added further information
to the methods section (lines 139-144, 155-162) and table 1 to clarify sample numbers in the
revised version of the manuscript.

2.) The authors also put a lot of influence on canopy cover dictating cover of biocrusts but their abiotic
variables are likely influencing the canopy cover. These things should be addressed together or the abiotic setting should be controlled for when looking at canopy cover. The disturbance that is
mentioned, that is the reason for the development of biocrusts is not described.

- Thank you for this comment. We agree that abiotic variables are influencing the canopy cover
just as well. Nevertheless, abiotic factors were mainly used to explain biocrust covers as their
influence on the tree layer is investigated by other scientists in separate studies within the BEF
China consortium. In our opinion, widening the analysis would lead beyond the scope of this
study.
Nevertheless, we added further information on clear-cutting and pre-disturbance vegetation
(lines 131-137). Unfortunately, there is no further data on biocrust species before the setup of
the project. The potential natural vegetation of this region is a subtropical broadleaved forest
with dominating evergreen species, which was replaced by a commercial forest plantation in the
1980's. This plantation was then clear-cut in 2008 when the BEF-China experiment was
established (see methods 2.1, Bruelheide et al. 2014, Yang et al. 2013).

3.) Specific comments: Line 114: The hypotheses could be stated more clearly. (1) "Biocrusts ARE widely
developED (2) "The development of biocrust is influenced by BOTH the surrounding vegetation cover
AND THE soil and terrain attributes."

- Hypotheses were adapted according to your and reviewer #2's comments (lines 116-121)

Line 173- It is not clear is the analyses met the assumptions of ANOVA.

- The dataset was tested and met the assumptions for ANOVA. We added further information on
normality and multi-collinearity to this paragraph (lines 197-198).

Line 185- "Than" should be "then".

- Changed

Line 239- I thought that this paper was primarily about soil erosion and biocrusts but that it not clear
from the first paragraph of the discussion.

- The paper is structured along the three hypotheses (1) Biocrust extend under forest, (2)
Influence of vegetation, soil and terrain on biocrust development, (3) Impact of biocrusts on soil
erosion. Thus, we used the same order in the results and discussion part. The title is pointing
more on soil erosion, though, as this was the most important finding in our opinion.
Nevertheless, biocrust development and e.g. classified moss species are of high importance, too.

Line 240- It is confusing to state hypotheses by numbers but quickly paraphrasing the hypothesis would
make interpretation easier for the reader.

- We agree that paraphrasing the hypotheses in the discussion part enhances the
comprehensibility for the reader. Thus, we added short repetitions of the hypotheses at the
beginning of each discussion paragraph (lines 247-248, 265-266, 285-286, 320-321).

Line 241- Is there any pre-disturbance data? It is hard to understand the connection between
interspaces and disturbance without some description of the pre-disturbance structure of the
vegetation.

-  See comment above. Unfortunately, we do not have any data on biocrusts and biocrust
development derived before the establishment of the experiment in 2009-2010. The pre-
disturbance vegetation was a *Cunninghamia lanceolata* plantation (lines 132-133). The methods
section has been adapted accordingly (131-137).

Line 271- Cite Condon and Pyke 2016, who have been able to restore a great deal of moss cover very
quickly.

-  Thank you for pointing out further literature, which has been added. The work of Condon and
Pyke gives very valuable insights on moss development after vegetation disturbance and fits
well in our context (line 280).

Line 321- You would have a stronger close if you finished with the sentence that ends here. It's also
unclear given your findings if there is much of a need to restore biocrusts since you saw recovery of
bryophytes really quickly.

-  We completely agree and the last two sentences have been removed from the manuscript.
Furthermore, we also believe that the need to restore biocrusts in mesic environments where
biocrusts establish quickly is less evident, than in arid environments (lines 281-282).
Nevertheless, we assume that biocrusts play an important role in young tree plantations, as high
soil losses can already occur in single heavy rainfall events, causing high damage when soil
surfaces are not sufficiently covered. Additionally, there is not much literature about biocrust
recovery in temperate climates and under forest plantations. The end of the paragraph has been
adapted accordingly (lines 276-282).

Line 335- You should remind the reader here of your scale as this likely influenced the effects of soil
attributes.

-  We added further information about scale and improved the manuscript as suggested by the
reviewer (line 366).

The authors need to work on the storyline of the paper as well.

-  We generally improve the writing and storyline throughout the manuscript and changed major
parts of the introduction and discussion.

[revised manuscript text omitted]

---

## Author Response (AR2)

**Response to reviewer comments on manuscript bg-2017-99: "Bryophyte-dominated biological soil crusts mitigate soil erosion in an early successional Chinese subtropical forest" – Minor revisions**

Dear Reviewers, dear Associate Editor,

Thank you again for taking the time to revise the manuscript and helping us to improve our work.

We now prepared a revised version where we account for all points raised by the reviewers, as described below. We show the reviewers' comments in grey text, while our responses are formatted as standard text. **Line indications refer to the revised manuscript without marked changes.**

Anonymous referee #2, report #1:

Dear Authors,

thank you very much for the in-depth revision of your manuscript, which reads much better as it stands now. From my side, there are two minor comments left, that you should be corrected before the manuscript can be published.

1) This is still about the classification of the crusts in the methods section. You corrected sentence reads as follows:" Biocrusts were described in the field based on appearance, functional groups and species composition and biocrust types determined based on the dominating autotrophic component". Unfortunately it is still not clear to me what you determination criteria were. Appearance, functional groups, species comopsition or dominating autotrophic component? If this was the dominating autotrophic component, what the threshold to declare one organism dominant? Was this based on coverage then?

- We now understand that this point is not yet clear to every reader. Thank you again for pointing that out.
  In the field, we first described species within the biocrusts in ROPs by appearance and functional groups, as far as possible. Then, we classified the general crust type per ROP with the help of the dominating autotrophic component (cyanobacteria, algae, moos, liverwort etc.) and this classification indeed was based on coverage of the soil surface (majority within one ROP).
  We apologize for still not being clear in our explanation and changed the text passage according to the reviewers' comments (lines 141-143).

2) In table 1 all mean values should be given with standard deviation.

- Standard deviations were added to table 1 (page 27-28).

Anonymous referee #4, report #2:

Most studies concerning biocrusts have been conducted in dryland regions, while literature addressing biocrust topics in humid areas is very scarce. This study analyses the influence of different biotic and abiotic factors on biocrust cover and the effect of biocrusts on erosion in a humid area in China, reason why I think the results presented in this manuscript are novel and interesting. I also consider the authors have adequately addressed most of the comments previously made by reviewers.

I only have some minor comments:

L68-71. I do not think biocrusts compete with plants for light or nutrients. As said in previous lines, biocrusts appear in the areas where plants are not able to grow. As soon as light and water conditions permit, soils are occupied by vegetation. So, I would delete the sentence "In direct competition with phanerogamic plants" and would slightly change the arguments included in this paragraph.

- Thank you very much for this thought-provoking impulse. We agree that biocrusts often appear in (arid) areas, where phanerogamic plants are not able to grow. Nevertheless, foresters removed the tree and shrub vegetation on our particular study site artificially. Those plants are very well able to grow again and continuously do so next to pioneer biocrust species. In our opinion, there is a competition for light in particular, which will finally limit the development of biocrusts. Competition between cryptogams and vascular plants have also been observed in other temperate environments (see e.g. Büdel et al. 2014, Biodiversity and Conservation).
Therefore, we rather tend to keep the sentence as it is. Nevertheless, we could attenuate the phrasing, if the editor sees the need for changes too.

L81-83, L91, L107-109. All these assumptions are indeed the hypotheses of this study, so they should be moved to the end of the Introduction where the different hypotheses are stated.

- You are right; those assumptions are actually recapitulating the 2$^{nd}$, 3$^{rd}$ and 4$^{th}$ paragraph, respectively, which are in each case substantiating one of the three hypotheses. We slightly changed the order of the 3$^{rd}$ paragraph/2$^{nd}$ hypothesis, to better point that out (lines 89-90, 95-96).
If you do not wish to have those recapitulations in the introduction, we would suggest to delete them from the manuscript, rather than to move them to the end of the introduction, where the hypotheses are stated separately.

L129-130. Please, add also the potential evapotranspiration in the area. It is also important to include in this section the soil texture as it is a very important factor influencing biocrust cover and runoff and erosion rates.

- Unfortunately, there is no data on evapotranspiration available now. Nevertheless, we agree with you about the importance of particles sizes and included information on soil texture (lines 128-129). In relation to the wide distribution of sand and silt stones, the

particle size distribution on both experimental sites were quite similar having loam as the main texture class (see Scholten et al. 2017; Journal of Plant Ecology).

L211-212. Please, include in the text the average moss cover of the plots in the year 2013.

- Further information on mean cover in 2013 was added (lines 214-215).

L270-271. At the end of the study, the authors observe "indications that biocrust cover may start to be pushed back ". It seems that after a certain threshold in LAI or crown cover, increasing shade provided by denser vegetation has a negative effect on biocrust cover. Could the authors provide a value of LAI/ crown cover/vegetation cover at which the shift to decreased biocrust cover is observed?

- Thank you for this remark. We completely agree that there is some kind of turning point or threshold in vegetation/canopy cover regarding biocrust development. We now included information on tree vegetation cover in summer 2016 (lines 273-274).
Unfortunately, as this observation was made at the end of the study, no further investigations or data on biocrust development could be included for 2016, so conclusions have to stay speculative. Furthermore, the LAI is very heterogonous between different tree species and in our opinion, further conclusions should be drawn based on the tree species level.

L322-324. The discussion about the effects of moss biocrusts on erosion should be improved by including more comparisons with other studies that have also reported erosivity/erosion rates in moss-dominated crusts. Some examples:

Knapen et al (2007) Effects of microbiotic crusts under cropland in temperate environments on soil erodibility during concentrated flow. Earth Surface Processes and Landforms.

Zhao et al (2014) Response of biological soil crusts to raindrop erosivity and underlying influences in the hilly Loess Plateau region, China. Biodiversity and Conservation.

Zhao and Xu (2013). Runoff and Soil Loss from Revegetated Grasslands in the Hilly Loess Plateau Region, China: Influence of Biocrust Patches and Plant Canopies. Journal of Hydrologic Engineering.

Bu et al (2015) The Combined Effects of Moss-Dominated Biocrusts and Vegetation on Erosion and Soil Moisture and Implications for Disturbance on the Loess Plateau, China. Plos One.

- Thank you for pointing out further literature, which we included in the manuscript to widen the discussion part on soil erosion rates in drylands (lines 329-330).

L336-337. Some studies in semiarid and temperate environments have reported an increase in soil hydrophobicity with the presence of later successional biocrusts (e.g. moss-dominated biocrusts), which could have a positive effect on runoff. See for an example "Drahorad et al. (2013) Ongoing succession of

biological soil crusts increases water repellency—a case study on Arenosols in Sekule, Slovakia". Discuss very briefly in the text (and as compared with the results found in this area) the possible opposite (positive) effect of mosses on runoff in some areas due to an increase in soil hydropohobicity.

- Indeed, water repellency might also play an important role to mitigate soil loss in this system and it would be very interesting to conduct further research on this topic in forest environments. We included the mentioned literature by Drahorad et al. and added a brief explanation to the discussion part of our manuscript (lines 352-355).

Anonymous referee #5, report #3:

The topic of this work is very interesting: it is revealed the significant role of a "not always recognized player (bryophytes) in forests" reducing soil erosion during the restoration of a subtropical forest after disturbance. This work also contributes to the knowledge of biocrusts succession in a subtropical climate. As the succession dynamic and roles of biocrusts in humid forests are very scarce, any proper contribution to this knowledge will help to predict the effects of human disturbances and to advance on modelling of the dynamic of these ecosystems under a global change context.

However, I am concerned on two aspects of the paper:

1.- In its present form, biocrusts succession analysis does not actually depend on the erosion/runoff study and furthermore the effects of different stages of succession on erosion is not presented. In fact, both analyses are quite independent to each other, so why not focus the paper on one of the topic, going deeply on it. If not, I would recommend increasing the connexion between both issues, providing data analysis linking them.

- We agree, that additional data analyses might reveal more and interesting new insights. We do also plan to conduct further studies on biocrusts and erosion under forest in general, as well as in this specific Chinese forest experiment. Thus, we thank the reviewer very much for his comments and will consider them for our future work.

2.- I am also concerned on the statements about soil properties effects on biocrust development. On the one hand authors do not indicate soil depth sampling, which is very important because biocrusts have been demonstrated to be influenced by upper layers of the soil. Also it is crucial to indicate when soils were sampled, because biocrusts are also affecting upper soil properties, so depending on the sampling moment we can speak on the influence of soils on biocrusts cover or on the influence of biocrusts on soils properties. On the other hand, only 3 soil properties are analysed, without explaining why these properties and no others were selected, and with them authors conclude that soil attributes did not have effect. Maybe other soil properties could affect biocrusts development such as texture, pools of SOC, CEC or even soil sealing.... With the mentioned limitations it is very difficult to evaluate the influence of soil.

- We agree that the influence of soil parameters on biocrust development are not completely covered by this study. Due to limited time and financial budgets we were only able to test a selection of parameters. The three mentioned properties seemed the most appropriate choice to us and we believe that they are suitable to explain major effects on biocrust

development (pH) and biocrusts and soil erosion (SOM, bulk soil density) together. Nevertheless, we fully agree that an overall evaluation of the influence of soils cannot be done based on this data set.
Thus, we attenuated the phrasing and explained the limited interpretation possibilities (lines 118, 288-289, 322-323). Moreover, information on soil sampling depth and time were added (lines 166-167).

Some minor points:

Although I realize that the idea of the limited number of studies on biocrusts from subtropical forest environments is key, however it is somewhat repetitive in the introduction and discussion. For introduction I recommend more emphasis on the impacts of filling this knowledge gap.

- Thank you for this comment. We tried to omit redundant repetitions and adapted the introduction according to your comment (lines 78-79, 89-90).

This study aims to investigate the development of biocrust cover in an early successional subtropical forest ecosystem after human disturbance and the impact of those biocrusts on soil erosion. I think the reader first impression is that you mean the impact of biocrust development on erosion, however the impacts only refer to a given moment, of the development, I recommend justifying the interest of choosing this moment.

- This study first aimed on soil erosion in an early successional subtropical tree plantation (for more detailed information please see Seitz et al. 2016, Goebes et al. 2015). While setting up the study, we already saw that biocrusts are developing widely and might also have a substantial influence on our measurements. We then immediately started recording the biocrust development within our ROPs.
We agree with you, that the impact of biocrusts on soil erosion might be limited to a given moment in time, most likely during the first years of the plantation with young tree ages and only little leaf litter on the forest floor. Thus, we decided to measure erosion rates after the first three years after tree planting (lines 157-159).

Methods:

2.2 Within field methods, in my opinion it should be better clarify if ROPs for biocrust cover measurements coincide with those for soil erosion measurements. It is not very clear the experimental design description, especially concerning plots types and measurements, I think it may be improved.

- Biocrust cover measurements and biocrust type classification took place in the same ROPs, in which we also measured sediment delivery and runoff. We clarified this in lines 143-145, 157-159).

Also it should be indicated if soil analysis were done at the start of the experiment and it is very important to indicate the sampling depth

- Soil analyses were done in 2013 and at the same time when the erosion and extended biocrust cover determination took place. Soil sampling depth was 0-5 cm and further information was added to the methods section (lines 166-167).

2.3, I recommend to specify at least the resolution of the DEM, to have an idea of its capability to capture topographic changes.

- DEM resolution was 5 x 5 m and information was added to the methods section (lines 172).

Table 1: Sediment delivery, runoff and rainfall should be expressed in g/m2 and mm respectively or at least the source area should be indicated in the caption (0.16m2?)

- You are right, sediment delivery, runoff and rainfall should be given in g m$^{-2}$ or l m$^{-2}$ (mm). Please apologize this shortcoming; the table was modified according to your comment (page 27-28).

Additionally, we added acknowledgments (line 404), as we have successfully applied for an open access publication grant at 
[revised manuscript text omitted]

---

## Author Response (AR3)

1    **Response to final editor comments on manuscript bg-2017-99:**

2    **"Bryophyte-dominated biological soil crusts mitigate soil erosion in an early successional Chinese**
3    **subtropical forest"**

4

5    Dear Associate Editor,

6    Thank you for your final comments and for accepting our paper for publication in BGS.

7    We agree with your comments, modified the introduction and removed the closing sentences starting
8    with "We assume…" (lines 78-81, 89-94, 102-104). Additionally, we rephrased the "personal
9    observation" in lines 353-354 as indicated, but also in lines 333-335 talking about *Pogonatum* and
10   *Atrichum*.

11   Furthermore, we added a very recent publication about the methodology of the BEF China experiment
12   (Trogisch et al. 2017).

13   Thank you again and best regards,

14   Steffen Seitz